# Smartphone Screen Integrated Optical Breathalyzer

**DOI:** 10.3390/s21124076

**Published:** 2021-06-13

**Authors:** Jerome Lapointe, Hélène-Sarah Bécotte-Boutin, Stéphane Gagnon, Simon Levasseur, Philippe Labranche, Marc D’Auteuil, Manel Abdellatif, Ming-Jun Li, Réal Vallée

**Affiliations:** 1Centre d’Optique, Photonique et Laser (COPL), 2375 Rue de la Terrasse, Université Laval, Québec, QC G1V 0A6, Canada; stephan.gagnon@copl.ulaval.ca (S.G.); simon.levasseur@copl.ulaval.ca (S.L.); philippe.labranche.1@ulaval.ca (P.L.); marc.dauteuil@copl.ulaval.ca (M.D.); rvallee@copl.ulaval.ca (R.V.); 2Groupe de Recherche Indépendant en Science des Données et des Décisions (GRISDD), 633 Ave. Des Oblats, Québec, QC G1N 1W1, Canada; helene@grisdd.com; 3Polytechnique Montreal, C.P. 6079, Succ. Centre-Ville, Montreal, QC H3C 3A7, Canada; manel.abdellatif@gmail.com; 4Corning Incorporated, SP-AR-02-5, Corning, NY 14831, USA; LiM@Corning.com

**Keywords:** breathalyzer, wearable, sensors, breath analysis device, health, mobile screen, alcohol, ethanol, smartphone, multimedia screen

## Abstract

One third of fatal car accidents and so many tragedies are due to alcohol abuse. These sad numbers could be mitigated if everyone had access to a breathalyzer anytime and anywhere. Having a breathalyzer built into a phone or wearable technology could be the way to get around reluctance to carry a separate device. With this goal, we propose an inexpensive breathalyzer that could be integrated in the screens of mobile devices. Our technology is based on the evaporation rate of the fog produced by the breath on the phone screen, which increases with increasing breath alcohol content. The device simply uses a photodiode placed on the side of the screen to measure the signature of the scattered light intensity from the phone display that is guided through the stress layer of the Gorilla glass screen. A part of the display light is coupled to the stress layer via the evanescent field induced at the edge of the breath microdroplets. We demonstrate that the intensity signature measured at the detector can be linked to blood alcohol content. We fabricated a prototype in a smartphone case powered by the phone’s battery, controlled by an application installed on the smartphone, and tested it in real-world environments. Limitations and future work toward a fully operational device are discussed.

## 1. Introduction

Extensive research has been conducted on portable breathalyzers [1,2,3,4,5,6,7,8]. Thousands of articles and patents have been published about the consumption of alcohol in recent years, and the numbers are increasing exponentially. The scale of scientific investments is a revealing response to the numerous tragic accidents linked to alcohol abuse. At least one impaired driving incident per 500 people is reported annually, and about 1 in 3 fatally injured drivers in North America was found to have a blood alcohol content (BAC) in excess of 0.08% in recent decades [9,10]. This represents over 10,000 deaths per year linked to impaired driving in the United States alone. Alcohol abuse is therefore understandably an important social concern.

In order to know if they can drive, people mostly rely on how they feel or the number of drinks they consumed. The problem with this method is that first, the BAC is influenced by the amount of food, the gender and the body mass of a person [11], and second, the intensity of the symptoms (nausea, slurred speech, lack of coordination) varies from person to person, depending of their drinking habits [12]. Therefore, two individuals with the same BAC can have a very different perception of their level of intoxication. Besides using a breathalyzer or blood sampling, there is no heuristic or approximation that can help a person figure out by themselves whether they are legally allowed to drive or not.

As of now, two of the most effective measures that prevent drinking and driving-related fatalities and injuries are roadblocks and ignition interlock devices [13]. One might think that the surge of availability of portable breathalyzers would help improve the situation. Unfortunately, this hypothesis has been proven wrong. Their use is not popular, as people are simply not interested in carrying such a device [14]. On the other hand, smartphones and wearables are widely used. Having a breathalyzer built into a phone or a wearable could be the way to get around the reluctance to carry a separate device, and may have a real impact on impaired driving or other alcohol related-incidents.

For these reasons, any milestone toward an inexpensive breathalyzer built into a phone or any wearable is of great importance. In this article, we propose a technology that could meet this demand. Our approach only requires a photodiode placed on the side of the glass screen of any mobile device such as smartphones or smartwatches. The photodiode measures the optical signature of the light from the display that varies according to the evaporation rate of the water vapor when a user breathes on the screen. The evaporation rate varies for different BACs. In this paper, the principle of our technology is first explained. Then, results using our prototype in the laboratory and in real-world environments are presented. Finally, the ambient and breathing conditions that affect the BAC measurements are studied, and future work to increase the breathalyzer precision is discussed.

## 2. Breathalyzer Principle

Our optical breathalyzer is based on the evaporation rate of water vapor from the breath. As shown in Figure 1a, when a person fogs a glass window (e.g., a smartphone screen) with their breath, thousands of microdroplets are formed on the glass surface. The evaporation rate of the microdroplets depends on the alcohol concentration in the breath [15], which is linked to the BAC [1,2,3,4,5,6,7,8]. On a dry screen, light from the smartphone display crosses the glass screen, and only a small part is reflected (Fresnel reflection) back to the display; see Figure 1b. As shown in Figure 1c, when a droplet is formed on the glass screen, part of the light is guided toward the edge of the glass screen (as in an optical fiber). Indeed, the edge of the droplet allows a strong oblique reflection (total internal reflection) which can be coupled into the planar waveguide formed by the dense antiscratch stress layer. This stress layer is found in multimedia device screens made of toughened glass, such as Corning Gorilla^®^ and AGC Inc. (Tokyo, Japan) Dragontrail^®^ glasses. Note that ray optics is insufficient to treat this coupling behavior with the planar waveguide. Using wave optics permits us to see that light sources near the waveguide interface can interact with the evanescent field tails of the waveguide modes, and hence, can transfer some of their power to them [16,17]. Therefore, only the light reflected from the very edge of the water droplets can be efficiently coupled to the planar waveguide at the surface of the glass screen. This is the key principle of our optical breathalyzer. Recent work has demonstrated the integration of invisible photonic devices and sensors in smartphone screens using laser writing of optical waveguides [18,19,20,21]. However, the complexity of laser writing added to the use of an external light source increase the cost and the mass production complexity. In our approach, the use of the display light already present and the stress layer planar waveguide greatly simplify integration into smartphones.

Figure 1d is a photograph of the side edge of the glass screen when the phone displays a white image at maximum brightness while the glass screen is fogged by the breath. The illuminated line comes from the microdroplets, since the image is entirely dark when there are no microdroplets (i.e., when the screen is dry). Using a photodiode placed at the edge of the glass screen, the evolution of the light intensity curve over time can be linked to the evaporation rate of the microdroplets. The aim of this article is to demonstrate that this optical principle can be used to monitor BAC by breathing on a smartphone screen.

## 3. Breathalyzer Prototype

At this stage, the technology is not fully integrated into the smartphone, for obvious reasons. Nevertheless, since the only required component that is not already part of a smartphone is the photodiode placed on the edge of the screen, we believe that the technology could be easily integrated. As shown in Figure 2a, our prototype acts as a protective case, which is also ubiquitous among smartphone users. It is fabricated using a Corning Gorilla^®^ glass screen, as found on most smartphones. A 7.5 cm × 13.5 cm piece has been cut to fit over a standard smartphone. This glass screen is placed on the top of the smartphone screen and does not affect its functions. The side edge of the Gorilla glass screen was polished to optical quality (down to a 0.5 µm grid) prior to installation of the photodiode. Figure 2b schematizes how the photodiode has been installed on the side of the glass screen to optimize the collection of light from the microdroplets and minimize noise.

First, a UV curing glue (Norland Optical Adhesive NOA63) with a refractive index that matches that of the glass to minimize the Fresnel reflection was used (1) to attach the photodiode (2). The silicon photodiode used (Advanced Photonix PDB-C612-2) had a thickness of 360 µm with an active area of 3.91 mm × 17.55 mm. Then, a thin layer of UV curing glue with a low refractive index *n* = 1.4 was applied on the glass surface, as shown in Figure 2b (3), to maintain the guiding property of the stress layer. A strong epoxy glue was then applied (4), covering the entire photodiode to increase its shock resistance. Finally, a very opaque black paint was applied (5) to minimize the noise from the ambient light. To measure the voltage generated by the photodiode, a voltmeter chip (2 × 4.5 × 1.16 cm^3^) from Yoctopuce (Yocto-milliVolt-Rx) was connected to the photodiode and the smartphone USB port (see Figure 2c). Finally, as shown in Figure 2d, an application was programmed to guide the user in running the breath test. In the first pane, the three dots at the top right were used to enter the ambient temperature and humidity. When the user touches the “Breath analyzer” rectangle, the next pane on the right is displayed and the user simply needs to follow the steps until the result of his BAC is displayed, as shown in Figure 2a. Of course, there is considerable room for improvement. The entire prototype uses the phone’s battery to operate.

## 4. Results and Discussion

### 4.1. Laboratory Tests

Before testing our prototype in a real-world environment, tests were conducted in the laboratory in a controlled environment (class 100,000 clean room). A volunteer was asked to breathe several times on the prototype for about two seconds under the same conditions to the best of their ability, before and after drinking alcohol. To compare our prototype with an accurate BAC value of the volunteer, four breath tests (two just before and two just after using our prototype) were carried out using a breathalyzer (model APC-90 from Alco Prevention Canada Inc., Laval, Canada) approved by the U.S. Food and Drug Administration (FDA). Figure 3a shows typical curves of the evolution of the light intensity over time, as measured by our prototype, for different BACs. The light intensity increased while the volunteer was breathing on the screen. When they stopped breathing, the light intensity decreased to zero (when the screen was completely dry). For each curve, the time corresponding to the moment when the evaporation was complete, which is the moment when the light intensity reached its lowest value for the first time, was set to *t* = 0 s.

A water droplet can evaporate twice as fast with 20% of alcohol compared to without alcohol, depending on humidity and ambient temperature [15]. However, the rate of evaporation is not linear based upon the alcohol concentration due to surface tension effects. Note that breath vapor can contain an alcohol concentration of up to 14%, which is equivalent to a BAC of 0.4 g/100 mL. At higher concentrations, coma or death is likely [22]. In addition, the proximity of the microdroplets when the screen is fogged further complicates the dynamics of evaporation. Indeed, the more droplets that are present, the more the humidity of the ambient environment increases, which decreases the rate of evaporation. This is the reason why the middle of a surface remains fogged longer than the microdroplets located at the edges, as shown in Figure 1a. It is therefore very difficult to analytically study the evaporation, and thus the light intensity curve measured by the detector of our breathalyzer presented in Figure 2. A very simple parameter to analyze is the time it takes for the microdroplets to evaporate completely. The time taken for the light intensity to go from 0.12 mV to 0 (the moment when the light intensity reaches its lowest value for the first time) was used. Figure 3b shows the evaporation time according to BAC, as measured with the APC-90 commercial breathalyzer.

The coefficient of determination obtained for the *n* = 57 measurements compared to the best polynomial fit (dotted curve) is R^2^ = 0.824, which demonstrates a good correlation between the measurements of our prototype and the actual BAC. The standard deviation of the BAC measured with the prototype is *σ* = 0.037 g/100 mL, which is far from the precision of the APC-90, i.e., 0.005 g/100 mL. A device with precision weaker than 0.02 g/100 mL could probably only be useful as alcohol interlock, for detecting whether the user is under the influence of alcohol or not. Note that this latter function is becoming commonplace for access control to restricted areas, such as nuclear power plants, process industry, and mining premises [6].

### 4.2. Real-World Tests

The ultimate goal is to design an accurate breathalyzer which is functional in real-world environments. To identify all the parameters that affect the measurement of our optical breathalyzer, tests were carried out during festive events; 140 measurements were carried out on 36 volunteers during three festive evenings, two in a house (Figure 4, black markers) and the other in an indoor public place (Figure 4, blue markers). Figure 4 shows the evaporation time measured with our prototype compared to the BAC measured with the APC-90. By analyzing the 140 evaporation curves over time (similar to those in Figure 3), several parameters and conditions affecting the measurements were identified. The following two sections (Section 4.2.1. and Section 4.2.2.) describe the parameters that were taken into account in our postprocessed results (Section 4.2.3). Other considerations and interferences that were excluded from this work are also discussed in Section 4.3.

#### 4.2.1. Breathing Condition

The opening of the mouth, the distance to the screen, and the strength and duration of the breath are parameters that can influence the number of microdroplets as well as their combination to produce larger droplets, thus affecting the dynamics of evaporation. Therefore, a mechanism capable of normalizing measurements, as implemented in commercially available breathalyzers, could be used. However, the original goal of not having to carry additional items would not be achieved.

The number of microdroplets directly affects the maximum measured light intensity as well as the evaporation time. For example, one of the volunteers was not able to breathe strongly and for long enough to fog an area large enough to obtain a maximum light intensity comparable to a normal measurement. Note that the volunteer was also unable to operate the APC-90 due to their shortness of breath. Under such circumstances, a message on the screen would explain that a longer breath is required. Nevertheless, in addition to the maximum light intensity, information from the signature of the curve during the breath can be obtained. The duration of the breath can be measured between the point where the curve starts to increase and the point where the curve starts to decrease exponentially. Moreover, the initial slope provides information on the strength of the breath. Considering these data, atypical situations could be identified and compared. The situations where the initial slope, the maximum intensity or the duration of the breath is the lowest are denoted by black (house) and blue (public place) empty squares in Figure 4. Clearly, these conditions have the effect of reducing the evaporation time.

When there is a large amount of water coming from the breath reaching the screen, the microdroplets can combine. Since only the edges of the droplets allow light to reach the detector, a large droplet produces a smaller light intensity than several small droplets measured at the detector. This situation is easily detectable on the curve of light intensity over time. When the light intensity decreases during evaporation, there is a moment when the light intensity stops decreasing (and can even increase). By analyzing images taken under a microscope over time, we confirmed that this moment corresponds to the separation of the microdroplets. Cases in which the microdroplets combined are represented by empty black triangles (house) and empty blue circles (public place) in Figure 4. Clearly, the combination of microdroplets has the effect of increasing the evaporation time.

#### 4.2.2. Ambient Conditions

Humidity, temperature, wind and ambient light can affect the measurements. Wind should be avoided, as it results in faster evaporation of the microdroplets. The breathalyzer should therefore be used indoors or in windless conditions. To maximize the accuracy, the ambient humidity and temperature should be known, which can be obtained with the sensors in modern smartphones [23,24], or entered by the user. Note that the temperature effect was not considered in this work. Although it was relatively similar for all our results (room temperature), the temperature should be precisely measured to increase the breathalyzer precision. Moreover, the screen should be properly cleaned before every test. Dirt on the screen produces a constant light noise on the detector (due to the same principle as with microdroplets). Other considerations regarding ambient conditions, which were not taken into account in the results of this article, are discussed in Section 4.3.

#### 4.2.3. Postprocessed Results

As shown in Figure 4, the breathalyzer is not accurate in real-world situations without postprocessing the data (R^2^ = 0.170 for *n* = 140). In fact, an evaporation time of over 10 s can be obtained from a BAC of 0.15, while an evaporation time of 5 s can be obtained from a sober person. However, considering a few simple parameters, the correlation can be greatly improved. By comparing the values in Figure 4 with the same ambient humidity, the coefficient of determination increases to R^2^ = 0.617 in the public place (blue markers) and R^2^ = 0.614 in the house (black markers). The humidity was 0.32 in the public place and 0.53 in the house, measured with a household hygrometer (Bios Weather). In addition, by simply discarding atypical data, as previously discussed (empty markers in Figure 4), the coefficient of determination increases to R^2^ = 0.793 in the public place and R^2^ = 0.723 in the house, which is evidencing the room for improvement that could be provided by appropriate postprocessing. We are currently investigating different approaches to address this issue.

### 4.3. Discussion

Periodic peaks are sometimes present in the light intensity curve. The intensity of these peaks is inversely proportional to the distance between the volunteer’s face and the screen. Indeed, smartphones are equipped with a source and a detector to measure the distance of objects in front of the screen [23,24]. It is this detector that turns off the display and the touch-screen functions when the user holds the phone near to their ear while engaging in a phone conversation. Since this source does not emit visible light, a simple filter placed in front of the breathalyzer detector would remove these peaks. The duration and intensity of these peaks provide information on the position of the volunteer’s face which could be used in the analysis of the evaporation curve. However, this information can most likely be acquired directly from the phone’s sensor.

We observed that ambient light produced noise in the measurements. In fact, ambient light can reach the detector due to the scattering in the glass screen. The breathalyzer response can be analyzed considering the light noise measured and compared with a reference with the same light noise. Restricting the use of the breathalyzer in the dark could also solve the problem. Adding an electronic filter that only keeps the display refresh rate frequency could discriminate it from other light sources such as sunlight. Adding an optical filter in front of the detector, allowing only the highest intensity of the display spectrum to pass, would also improve the results.

The accuracy of the breathalyzer could be improved in various ways. For example, a line-by-line illumination of the display could be performed prior to the breath monitoring, in order to identify defects like surface scratches and then subtract the corresponding noise. Line-by-line scans could also be performed during the breath test to determine the width of the area that is fogged by the person’s breath. The phone’s accelerometer data could be recorded to estimate the orientation of the phone, since the gravity affects the evaporation dynamics. Some phone’s sensors, such as the temperature sensor, might not be a reliable source of information under certain conditions. For example, if the processor was highly solicited or if the display was lit over a long period of time, its temperature might be overestimated, while after carrying the phone close to the body with the display facing away from the person, the temperature sensor is likely to be warmer than the display. Therefore, the software application could ask the user to put the smartphone on a horizontal surface for a few minutes prior to the breath test.

Under nonoptimal ambient conditions (judged by the user or detected by the smartphone), a measurement of the breathalyzer performed with the breath of a sober person could serve as a reference to calibrate the device and improve the accuracy of the next test.

## 5. Conclusions and Future Work

Our next step is to acquire a large data set of BAC optical signatures (as in Figure 3a) using our prototype in several festive events in order to train a DNN and use machine learning to obtain accurate BAC measurements. The ambient and breathing conditions discussed earlier will be used in a supervised training mode [25], in which the connection weights of the DNN are adjusted by minimizing the training loss function (difference between the actual DNN output and its desired output). Using this approach, we hope to develop a robust, ecological (no disposable parts or replacements of degraded chemical sensor elements), inexpensive and discreet wearable breathalyzer that can have a real impact on our society. Finally, our contactless technology could be an interesting approach for medical diagnoses and triage at emergency medical care facilities (or even in the street). It is common that certain medical conditions (head injury, stroke, heart attack, diabetes, or psychological illness) are mistaken for alcohol intoxication. Unfortunately, state-of-the-art breathalyzers require the active involvement of the patient, and the required expiratory volume and flow may be incompatible with the patient’s respiratory function [5,6].

Innovative applications using a smartphone as an operating medium are growing rapidly, demonstrating the demand for portable sensors [18,19,26,27,28,29,30,31,32]. The combination of these applications will hopefully someday make the smartphone a lab-in-a-pocket.

## Figures and Tables

**Figure 1 sensors-21-04076-f001:**
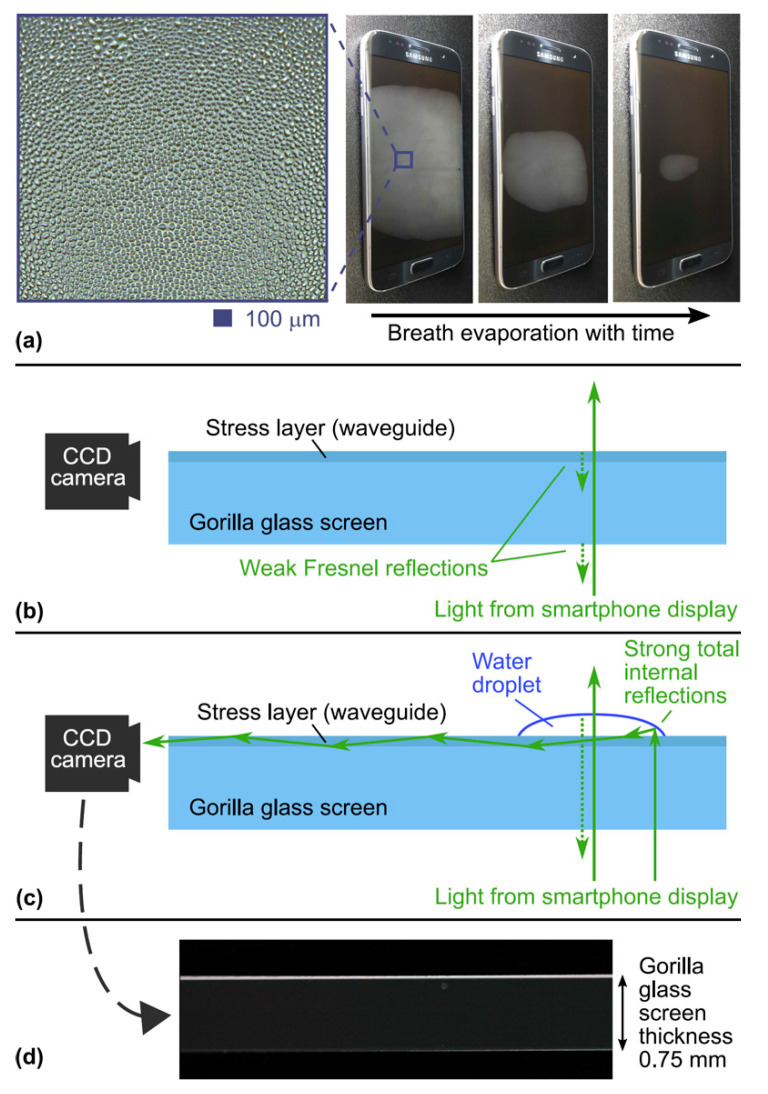
The optical breathalyzer principle. (**a**) Evaporation of the microdroplets when a person breathes on a glass screen. The inset is a zoom on the microdroplets. (**b**) The light from the smartphone display is not coupled to the planar waveguide on the surface of the glass screen. (**c**) With water droplets on the glass screen, the light from smartphone display is coupled to the planar waveguide, due to the strong oblique reflections at the edge of the droplets, and is guided to the side of the screen. (**d**) Photograph of the guided light in (**c**) using a CCD camera and a 10× objective lens.

**Figure 2 sensors-21-04076-f002:**
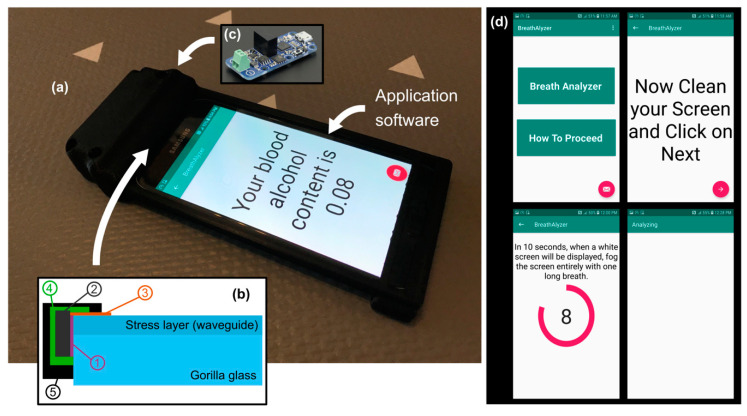
The optical breathalyzer prototype. (**a**) Photograph of our prototype. (**b**) Scheme of the photodiode installation on the side of the glass screen. (**c**) The electronics. (**d**) Screenshots of the application software steps before displaying the BAC result shown in (**a**).

**Figure 3 sensors-21-04076-f003:**
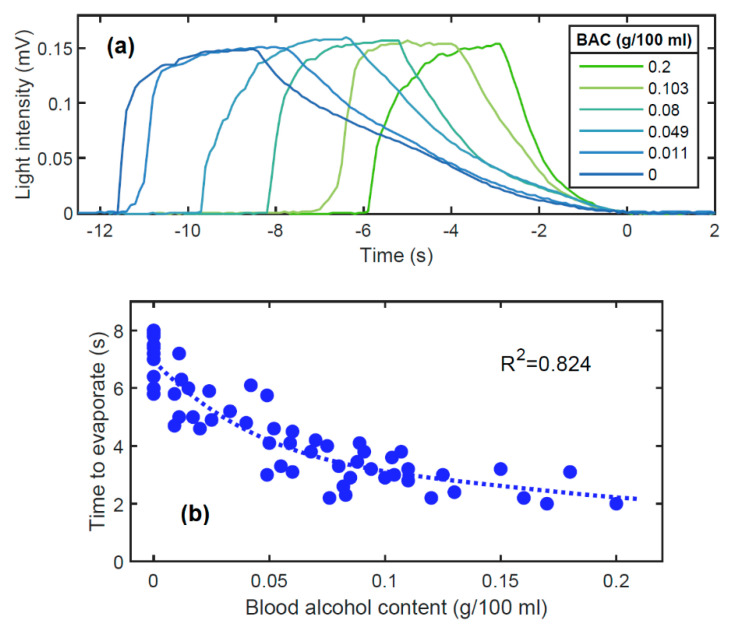
(**a**) Light intensity curves measured at the photodiode for different BACs. (**b**) Good correlation (R^2^ = 0.824) between the breath evaporation time and the BAC in laboratory environment.

**Figure 4 sensors-21-04076-f004:**
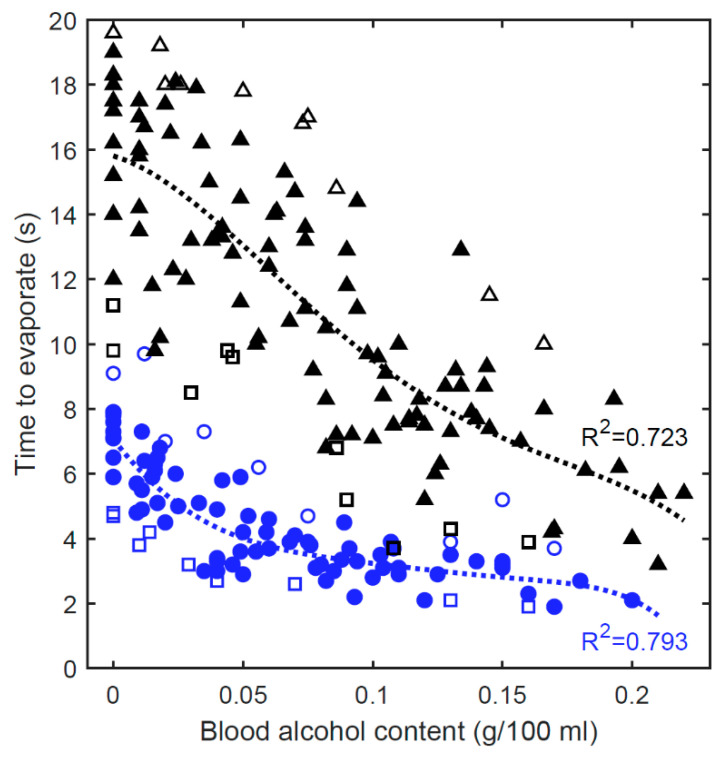
Utilization of our breathalyzer in a real-world environment. The weak correlation between the breath evaporation time and the BAC is greatly improved by considering a few measurable parameters. Blue markers: humidity of 0.32. Black markers: humidity of 0.53. Empty markers: detected anomalies.

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
