# Peer review of "Smartphone Screen Integrated Optical Breathalyzer"

_sensors, 2021, doi:10.3390/s21124076_

Round 1

Reviewer 1 Report

This paper proposed an inexpensive breathalyzer that could be integrated in the screen of mobile devices. The principle is based on the evaporation rate of the fog produced by the breath on the phone screen. This measurement method is innovative, and I have some questions.

  • Will the external conditions affect the results? For example, temperature.
  • May be adding some formulas in principle will be better.
  • Adding some related literature in introduction will be better.

Reviewer 2 Report

I really like the detailed presentation of the theoretical approach and your implementation of it. In addition to that, the experimental investigation of your approach is very thorough. Even though you identify your results of being a preliminary study, you have already collected a vast amount of data. And the many explanations for phenomenons, that interfere with your measurements, prove that you have a profound understanding of the subject.

However, when it comes to the presentation of your results, my praise is less unanimous. You conflate your results with the interferences during the measurements, proposed mitigations and an outlook of how you intend to continue this project. Due to this, the presentation falls short in many ways:

  • Topics like the combination of the microdroplets and your plan to use a DNN are mentioned multiple times. With this, the paper is longer than it needs to be.
  • It is necessary to read the paper very carefully to get an intuition about for which interferences with the measurement, you already have implemented or at least proposed a solution and which interferences are still not accounted for.
  • By drowning the presentation of your results in the many explanations of the phenomenons, that interfere with the measurements, it gets overlooked, how much data you have already collected and which conclusions you can already draw from this data.

Therefore, I recommend, that you separate the presentation of your results from the discussion of the phenomenons. And move the outlook for the further research to the end, so it is easier to describe, which investigations you plan to do and what topics you intend to leave for other researchers.

Another issue, that I see with this paper is, that it seems as if it could be significantly improved with a relatively small additional research effort. You leave a lot of open questions, for which you propose an easy way to answer them, without actually providing the answer:

  • I suggest, that you do include a filter to mitigate the interference with the distance sensor, so you know how much this interference impairs the measurement. (Also, I suggest, that you read the sensor's data directly rather than trying to estimate the distance between the person and the phone through the interference from the sensor's measurement signals, like you proposed).
  • I suggest, that you pursue your proposed approach of estimating whether a combination of microdroplets is happening, based on the sudden drop in measured light intensity.
  • I suggest, that you do synchronize your sensor with the display refresh rate (or even a deliberate strobing) in order to account for the influence of ambient light.
  • I suggest, that you do use a mouthpiece to control how wide the persons open their mouth during the test. This might not be suitable for the real-world implementation of the breathalyzer, but as a basic laboratory test, it is likely to help you separate and investigate the many phenomenons, that interfere with the measurement.

As another easy to implement extension, I propose to light the screen only partially during the measurements, in order to separate the influence of short breath and that of the combination of microdroplets. With a pattern similar to a scanner, that captures an image line-by-line, you should be able to determine the width of the area, that is fogged by the person's breath. And then you know better, whether a lower-than-expected light intensity comes from a small fogged area or the combination of the microdroplets.

You might also want to record the accelerometer data, in order to estimate the orientation of the phone. I expect, that gravity affects the combination of microdroplets and that it has a much smaller impact on the microdroplets than on the larger combined ones. Maybe, this helps you to determine if a combination of microdroplets took place or not.

I also expect some other interferences with the measurement, that you do not discuss. In the interest of keeping the paper concise, I recommend, that you do not go into too much detail on them, but maybe, you can make a list of requirements, that have to fulfilled, so the user can expect a reliable measurement.

  • scratches or even cracks in the glass will probably lead to an increased light noise. Does this make the measurement impossible or is there a way to calibrate or compensate such issues? (e.g. by not lighting up the display in the scratched areas)
  • There are other factors, that influence the chemical composition of the breath, such as a keto diet or health conditions like diabetes or liver and kidney issues. Are these influences significant enough to interfere with your measurement?
  • The phone's temperature sensor might not be a reliable source for estimating the temperature of the display. For example, if the display was lit over a long period of time, its temperature might be overestimated, while after carrying the phone close to the body with the display facing away from the person, the temperature sensor is likely to be warmer than the display.

Reviewer 3 Report

Dear Authors,

idea is very interesting. Also results are promissing. Referenes ae ok.. used properly and in the context.

Unfortunately, there is missing flow chart of SW solution.. and also missig modules owerviewe of HW solution.. So the idea is described at too abstract level. please try to add more details in these tow directions.
